
# Landslide risk zoning in Ruijin, Jiangxi, China

Xiaoting Zhou [1], Weicheng Wu [1*], Ziyu Lin [1], Guiliang Zhang [2], Renxiang Chen [2], Yong Song [2], Zhiling Wang [2], Tao Lang [2], Yaozu Qin [1], Penghui Ou [1], Wenchao Huangfu [1], Yang Zhang [1], Lifeng Xie [1], Xiaolan Huang [1], Xiao Fu [1], Jie Li [1], Jingheng Jiang [1], Ming Zhang [1], Yixuan Liu [1], Shanling Peng [1], Chongjian Shao [1], Yonghui Bai [1], Xiaofeng Zhang [3], Xiangtong Liu [4] and Wenheng Liu [1]

[1] Key Laboratory of Digital Lands and Resources and Faculty of Earth Sciences, East China University of Technology, Nanchang, 330013 Jiangxi, China

[2] 264 Geological Team of Jiangxi Nuclear Industry, Ganzhou, Jiangxi, China

[3] School of Geophysics and Measurement-Control Technology, East China University of Technology, Nanchang, 330013 Jiangxi, China

[4] Faculty of Geomatics, East China University of Technology, Nanchang, 330013 Jiangxi, China

* Correspondence: Wuwc030903@sina.com/ wuwch@ecut.edu.cn; Tel.: +86-18970847278

**Abstract:** Landslides are one of the major geohazards threatening human society. This study was aimed at conducting such a hazard risk prediction and zoning based on an efficient machine learning approach, Random Forest (RF), for Ruijin, Jiangxi, China. Multiple geospatial and geo-environmental data such as land cover, NDVI, landform, rainfall, stratigraphic lithology, proximity to faults, to roads and to rivers, depth of the weathered crust, etc., were utilized in this research. After pre-processing, including digitization, linear feature buffering and value assignment, 19 hazard-causative factors were eventually produced and converted into raster to constitute a 19-band geo-environmental dataset. 155 observed landslides that had truly taken places in the past 10 years were utilized to establish a vector layer. 70% of the disaster sites (points) were randomly selected to compose a training set (TS) and the remained 30% to form a validation set (VS). A number of non-risk samples were identified in low slope (< 1-3°) areas and also added to the TS and VS in the similar percentage. Then, RF-based classification algorithm was employed to model the probability of landslide occurrence using the above 19-band dataset as predictive variables and TS for training. After performance evaluation, the RF-based model was applied back to the integrated dataset to calculate the probability of the hazard occurrence in the whole study area. The predicted map was evaluated versus both TS and VS and found of high reliability in which the Overall Accuracy (OA) and Kappa Coefficient (KC) are 91.49% and 0.8299 respectively. In terms of the risk probability, the predicted map was further zoned into different risk grades to constitute landslide risk map. Modeling results also revealed the order of importance of the 19 causative factors, and the most important ones are the proximity to roads, slope, May-July rainfall, NDVI and elevation. We hence conclude that the RF algorithm is able to achieve the risk prediction with high accuracy and reliability, and this study may provide an operational methodology for geohazard risk mapping and assessment. The results of this study can serve as reference for the local authorities in prevention and early warning of landslide hazard.

**Keywords:** Landslide; Hazard factor quantification; Random Forest; Risk zoning

## 1. Introduction

Landslides are frequent natural disasters, which pose a serious threat to traffic, property and safety of people (Wu and Ai, 1995; Nadim et al., 2006; Assilzadeh et al., 2010; Froude and Petley, 2018). Petley (2012) stated that the damage caused by non-seismic landslides is severe around the


world, especially in Asia, and we used to underestimate the toll caused by this kind of disaster.
Ruijin County in Jiangxi, South China is such an area where landslide calamity constitutes a serious
threat and problem to human society. According to the 264-Geological Team (of the Geological
Bureau of Jiangxi Nuclear Industry), landslides have caused damage to 104 residential buildings,
and made 2319 people homeless in the past decades. Affected by landslide disaster, the construction
and use of the newly-built campus of No. 6 Middle School in Ruijin County have been suspended.
The uncertainty and suddenness of disaster constitute potential threats to human daily life (Nadim et
al., 2006; Froude and Petley, 2018). The recognition of potential landslide-prone areas acts as an
essential part in hazard early warning and aiding decision-makers in land use planning and resource
management, as well as reducing losses caused by disasters (Aleotti and Chowdhury, 1999; Wu et
al., 2016; Arabameri et al., 2020).
In the past decades, a number of studies about landslide risk prediction and assessment have
been conducted (Montgomery and Dietrich, 1994; Guzzetti et al., 1999; Aleotti and Chowdhury,
1999; Ayalew and Yamagishi, 2005; Ruffff and Czurda, 2008; Arabameri et al., 2017). These
studies have proposed a variety of landslide risk prediction and assessment methods, e.g.,
knowledge-based, physical, and data-driven methods (Corominas et al., 2014; Reichenbach et al.,
2018; Li et al., 2017). Actually, advantages and limitations exist in each approach, for example,
knowledge-based and physical methods are mostly intuitive but qualitative or half quantitative
(Corominas et al., 2014; Goetz et al., 2015; Li et al., 2017), while data-driven methods are
quantitative, yet require powerful computing capacity for big data processing. On the whole,
data-driven methods seem more promising for a higher prediction accuracy than other methods, and
thus shall be more suitable for landslide risk assessment in areas where there is insufficient
geotechnical data (Guzzetti et al., 1999; Corominas et al., 2014; Furlani and Ninfo, 2015; Li et al.,
2017; Zhu et al., 2019).
Owing to the heterogeneity in geological and environmental conditions, the scale and
mechanism of landslides may differ from one place to another (Cao et al., 2019). This makes the
hazard prediction complicated and requires a consideration of the hazard-causative factors as many
as possible while dealing with such risk assessment. Recently, remote sensing (RS) and Geographic
Information System (GIS) have been taking an significant part in the study of disaster risk zoning
(Grizer et al., 2001; Wu et al., 2003; Wang et al., 2005; Lai et al., 2019; Chang et al., 2020). RS
technique can provide not only multitemporal and time-series spatial information of large and even
inaccessible areas over span of decades but also timely pre- or post-hazard spatial data easily
(Youssef et al., 2009; Wasowski et al., 2015; Arabameri et al., 2020). Therefore, RS is an effective
tool for hazard monitoring and assessment. To be precise, satellite images can provide the
environmental factor layers (e.g., topography, land cover and anthropogenic activities) which can be
used for landslide risk prediction and modeling (Pradhan et al., 2010; Arabameri et al., 2020). The
other main intrinsic geological and meteorological hazard-inducing factors are also fundamental and
essential for this purpose.
In the past years, machine learning techniques including artificial intelligence and deep
learning have gained a momentum in geospatial big data processing. For example, data-driven
algorithms such as Support Vector Machines (SVM), Random Forests (RF), and Artificial Neural
Networks (ANN) have been well applied in land resource mapping (Wu et al., 2016), prediction of
soil salinity (Wu et al., 2018) and of ore mineralization in geological field, and shown a superior



performance to the traditional approaches (Huang, 2018; Qin et al., 2018; Achour and Pourghasemi,
2019; Dou et al., 2019; Sameen et al., 2020). Comparing with other machine learning approaches,
the RF algorithm has clear advantages, i.e., it does not require to normalize and discretize the data,
and is less sensitive to outliers and runs faster than SVM (Breiman, 2001; Wu et al., 2016; Zhang et
al., 2017). Landslide causative factors often present nonlinear relationships (Corominas et al., 2014;
Zhu et al., 2019). RF algorithm is able to catch such nonlinear features among the factors and also
to prevent overfitting (Breiman, 2001; Goetz et al., 2015; Arabameri et al., 2020).
In view of the reliable prediction result obtained from regression and classification with the RF
algorithm (Wu et al., 2016; Wu et al., 2018), the objective of this study is to employ this algorithm,
one of the data-driven methods, to model the landslide risk taking Ruijin County as an example. As
RF algorithm has been rarely applied to landslide study, one specific objective of this research is to
find out an operational RF-based approach for this kind of geohazard zoning and mapping.
**2. Data and Methods**
*2.1 The study area*
Ruijin County is located in the southeast of Jiangxi Province, China, extending from 115° 41'
10" to 116° 21' 49" E in longitude and from 25° 32 '15" to 26° 17' 45" N in latitude, covering an
area of about 2436 km$^2$ (Fig. 1). From the view of topography, the elevation of the study area varies
from 70 to 1211 m with a mean of 324 m while the slope from 0 to 65° with an average of 15°.
Hydrologically, the main rivers are Meijiang, Mianjiang and Jiubao Rivers running through the
study area as sub-tributaries of the Gongshui River watershed. The study area belongs to the
subtropical humid climate zone and is characterized by four distinct seasons, sufficient rainfall and
long frost-free period. Heavy rainfall often occurs from March to July, accounting for more than
50 % of the annual rainfall with amount of about 1663.5 mm, an average of the period from 1968 to
2017. The annual mean temperature is 21.54 ℃ and July is the hottest month of the year with a
mean temperature of 28.8 ℃.
The hot and humid weather leads to severe weathering of rock mass giving rise to formation of
a thick weathered crust in which most landslides take places. From the human side, artificial cutting
on slope for infrastructure construction (such as roads and highways) and housing development
provoke instability of the crust mass causing landslides.

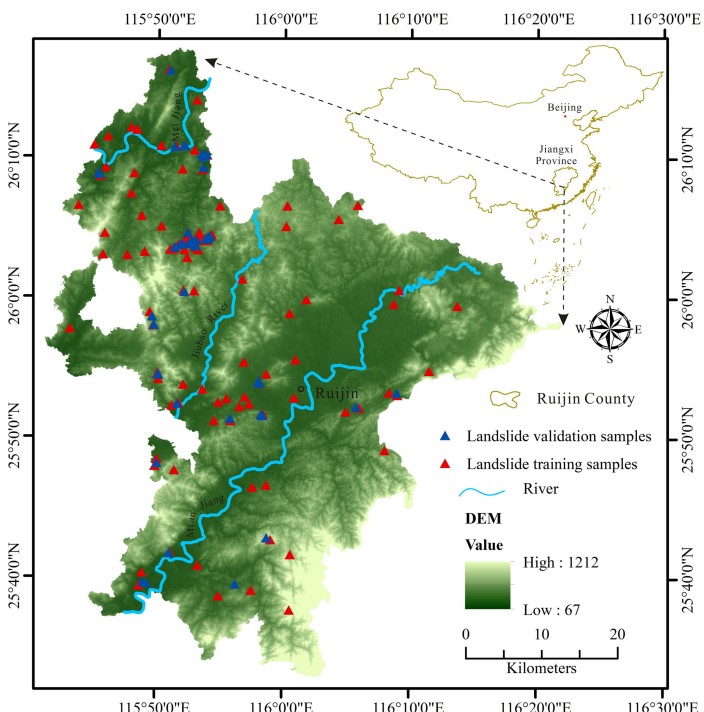

**Fig. 1** Location of the study area, Ruijin County, Jiangxi, China and location of the training and validation sites of landslides in the study area.

*2.2 Data and Processing Procedures*

For landslide risk assessment, it is unavoidable to deal with non-digital geo-environmental data such as geological strata, faults, rivers as they are essential for this purpose. It is hence necessary to convert them into digital and meaningful values so that they can be incorporated as quantitative variables for landslide risk modeling. The global methodological procedure includes data pre-processing, digitization, linear features buffering, rational numeric value assignment to descriptive factors and buffers, risk modeling and validation, and finally accuracy assessment. These procedures are presented in a flowchart shown in Fig. 2.



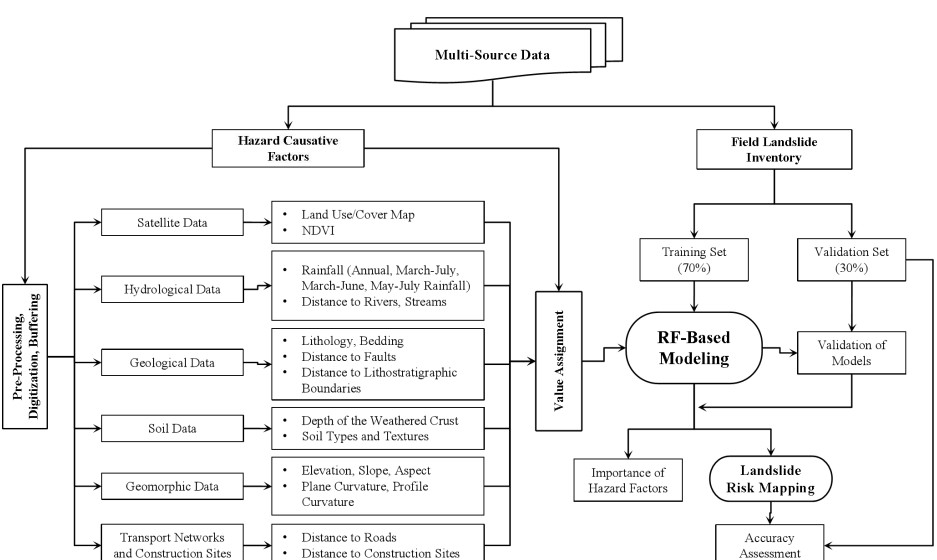

**Fig. 2** Methodological flowchart.


2.2.1 Geospatial and geo-environmental data
2.2.1.1 Satellite data
1) Landsat imagery: Landsat 5 TM images of October and November 2006-2010, and Landsat
8 OLI images dated May 2017 and Sept 2019 were obtained from the USGS data server
(https://glovis.usgs.gov). After atmospheric correction using COST model (Chavez, 1996; Wu,
2003 and Wu et al., 2013), Landsat 8 images were used for land cover mapping using the approach
proposed by Wu et al. (2016) and Landsat 5 data for deriving the averaged multiyear autumn NDVI
(Fig. 3a). In general, vegetation can help soil hold water content and improve its mechanical
properties through root system to stabilize slopes. Thus, landslides may arise more likely in
unvegetated areas rather than in vegetated ones (Montgomery et al., 2000; Reichenbach et al., 2014).
However, this will be completely different when slope is cut or excavated because of road
construction or housing development.
2) Very high resolution images: those are available on Google Earth as a complementary
source of ground-truth data. The road and river networks were also extracted from Google Earth
(Fig. 3b, 4d).
According to the principle of RF algorithm, we shall use two types of samples for modeling as
input variables: one is the locality of landslides that have taken places and the other is the stable
areas where landslides are unlikely to occur (Frattini et al., 2010; Depicker et al., 2020; Arabameri,
2020). Identified on Google Earth, the stable areas are places where slope is less than five degree,
e.g., water bodies, urban areas, and cultivated land. Landslides with an area greater than 900 m$^2$ (1
Landsat pixel) that had been overlooked during the field observation were also identified and
delineated on Google Earth.
2.2.1.2 Hydrological data
1) Rainfall: Rainfall is the main factor triggering landslides (Monsieurs et al., 2018; Depicker
et al., 2020). Depicker et al. (2020) stated that rainfall condition was the direct cause of many


shallow landslides. Daily rainfall data from Jan 2008 to Dec 2013 were obtained from 40
meteorological stations in Ruijin County and its adjacent areas. As landslides mainly occurred in
March to July, especially in June and July but without recorded occurrence time, our intention was
to investigate which months of rainfall may best reveal its role in landslide events. Thus, apart from
the mean annual rainfall, March-June rainfall, May-July rainfall and March-July rainfall of these six
years were also aggregated and gridded into raster with 30 m pixel size using Inverse Distance
Weighting (IDW) approach.

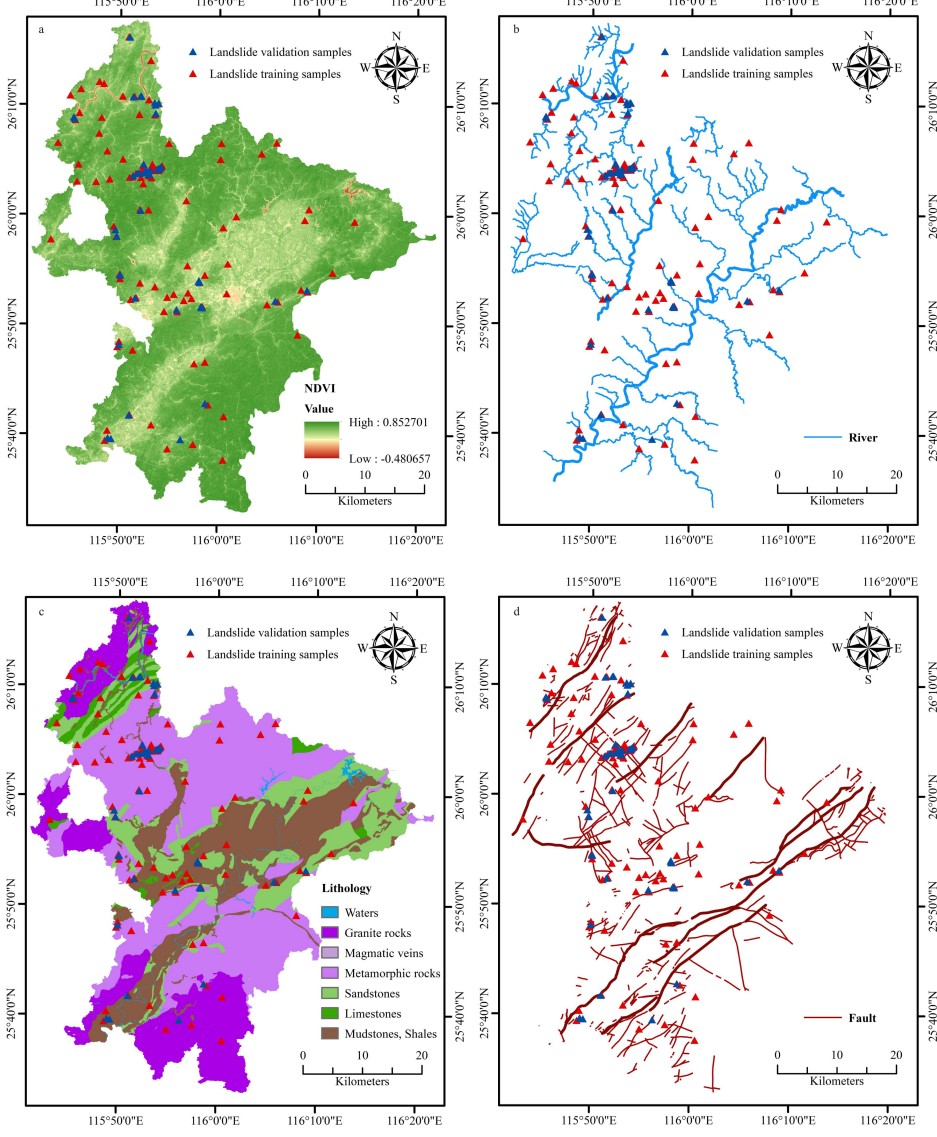

**Fig. 3** Landslide hazard factors: (a) NDVI, (b) Rivers, (c) Lithology, and (d) Faults.


2) River network: The influence of rivers on the occurrence of landslides is reflected by the
proximity to, or rather, the distance to them (Chen et al., 2018a; Cao et al., 2019; Arabameri et al.,
2019). Thus, the rivers were vectorized from Google Earth (Fig. 3b) and buffered into belts with an
interval of 30, 60, 90, 120 and 150 m, respectively, for streams and 60, 120, 180, 240 and 300 m
respectively for the main rivers. Then, these buffers were assigned values in terms of their
propensity or their importance in the event of landslide. For example, for the main rivers buffers of
0-60, 60-120, 120-180, 180-240 and 240-300 m were respectively assigned with 20, 15, 10, 5 and 1
while for streams, buffer zones of 0-30, 30-60, 60-90, 90-120 and 120-150 m respectively with 10,
8, 6 4 and 1. This implies that the closer to the river the higher propensity of landslide.
Finally, these buffers are converted to raster data with 30 m cell size using the "Polygon to
Raster" tool as proposed by Wu et al. (2018).
2.2.1.3 Geological and geomorphic data
1) Geological strata and formations: Geological strata were extracted from 1/50,000
Geological Map. Except for Ordovician, Silurian, Triassic and Tertiary, the strata of other
geological periods are mostly exposed. In terms of texture and composition, the lithology of
different strata in the study area can be divided into 113 classes. To facilitate the geohazard analysis,
these lithological classes were further aggregated into six main categories: (1) granitic rocks, (2)
magmatic veins, (3) metamorphic rocks, (4) sandstone, (5) limestone, (6) mudstone and shales as
shown in Fig. 3c. Just based on lithology and in absence of faults and joints, granitic massif would
possess the highest resistance while mudstone the lowest to landslides. Hence, from (1) to (6), the
propensity is likely to increase and they were respectively assigned values 1, 2, 3, 5, 7 and 10.
According to the field observation, landslide events occurred frequently on the boundaries
between two formations, especially, between the Quaternary sediments and other strata. Therefore,
the lithostratigraphic boundary factor was also obtained by buffering and rasterization, then added
to analyze the landslide risk.
2) Faults: This kind of geological structure has a prominent effect on the stability of rock mass
(Smets et al, 2016; Delvaux et al., 2017). There formed the spectacular thrust nappe structure which
was characterized by strong faulting activity in the study area. Such structure is accompanied with a
series of faults and folds, which tend to be the landslide-prone areas, e.g., the fragile belts related to
fold hinges, fracture zones and joints. As a matter of fact, the proximity to fault plays a role in such
hazard events, i.e., the closer to the fault, the higher risk of landslide may exist. For this reason, the
faults in the study area (Fig. 3d) were divided into three groups in terms of scale, i.e., big faults if
their length is > 10-20 km, medium faults if they are 2-10 km, and small faults if they are < 2 km.
The big faults were buffered into five zones of 0-120 m, 120-240 m, 240-360 m, 360-480 m and
480-600 m, and were respectively assigned with values of 20, 15, 10, 5 and 1. For the medium ones,
they were also buffered into five zones of 0-60 m, 160-120 m, 120-180 m, 180-240 m and 240-300
m with assigned values of 10, 8, 6, 4 and 1. The small faults were again buffered into five zones of
0-30 m, 30-60 m, 60-90 m, 90-120 m and 120-150 m and respectively assigned with 5, 4, 3, 2 and 1.
These fault buffers were gridded into raster layer of 30 m in resolution.
3) Depth of the weathered crust, soil type and texture: Weathering is the process converting
rocks into soils to constitute the weathered crust of our land surface. Landslides mostly take place in
this crust in which soil texture seems of significant impact on (Kitutu et al., 2009) and the
variability of soil types and depths of the crust play a part in the occurrence of such events (Fan et


al., 2016). Because different soil types and textures have different sand percentage, grain sizes and
porosity affecting the permeation of rain water. If liquidized by penetrated water, the crust bottom
(soil/rock interface) may serve as slip surface of landslides as friction and resistance from the
underlying rocks are reduced by this process. As soon as it has reached certain threshold, landslide
occurs. Thus, the crust depth, i.e., the depth of the slippery soil/rock interface, is a plausible
indicator of landslide volume and scale.

Here the data of soil types were obtained from the Bureau of Jiangxi Coal Geology and the
sand percentage (%), in which high sand percentage (low percentage of clay but high porosity)
seems favorable for permeation of rain water and provoking landslide event, was considered as an
indicator of soil contribution. Hence, soils with sand percentage > 40%, 30-40%, 20-30%, 10-12%,
5-10% and 0-5%, were respectively assigned with values of 10, 8, 6, 4, 2 and 1. At last soil
proneness map was converted into raster of 30 m resolution.

The depth data of the weathered crust were obtained from the field 1282 measurements. In
assumption that all the ridges have a crust of 0.5m in depth, these field observed depths were
interpolated using Kriging approach, then converted into raster layer of 30m resolution.

4) Geomorphic data: Slope (angle) is a key driver of landslides and a triggering angle threshold
of 28° - 38° was reported by Fan et al.(2016); at the same time, elevation, aspect, plane curvature
and profile curvature may also contribute to the occurrence of the hazards (Corominas et al., 2014;
Guzzetti et al., 2005; Galli et al., 2008; Pourghasemi and Kerle, 2016; Cao et al., 2019). The
ASTER GDEM data with a spatial resolution of 30 m, jointly developed by METI of Japan and
NASA of the United States, were obtained for Ruijin County from the Geospatial Data Cloud
(http://www.gscloud.cn/) and used to derive elevation, slope, aspect, plane and profile curvatures
(Fig. 1, Fig. 4a, 4b).
2.2.1.4 Land use/cover, transport system and construction sites
Using the classification approach proposed by Wu et al. (2016), land cover mapping was
achieved for Ruijin with an accuracy of 90.99%. The main land cover type is forests (54.25%),
followed by shrub/woodlands (29.33%), croplands (6.65%), artificial areas (urban, villages, road
and other infrastructures, 5.36%), barelands (1.45 %) and waters (1.41%) (Fig. 4c). Forests cover
hills and mountains, artificial areas and croplands are mainly distributed in lowlands (valleys) with
low slope. For risk modeling purpose, forest cover was considered of low proneness and assigned a
value of 1-2. On the contrary, unvegetated hilly slopes and barelands were regarded of high
propensity and assigned a value of 10, while zero-slope croplands, urban and water-bodies were
treated as non-risk (zero probability) areas. At the same time, NDVI can be used as an indicator of
vegetation greenness and abundance, indirectly representing the development degree of the root
system. For barelands, woodlands and forests, NDVI shall be a good indicator of propensity of
landslide.

Road construction is one of the important human activities leading to slope failure
(García-Rodríguez et al., 2008; Cao et al., 2019). Similarly, housing development along two sides
of the roads or on the brink of hills by cutting slopes constitutes also an important factor making the
slope massif unstable. The influence of road on landslide is also reflected by the distance to them
(Chen et al., 2018a; Cao et al., 2019; Arabameri et al., 2020). The road system (Fig. 4d) was also
assigned values the same as was done for rivers and faults.

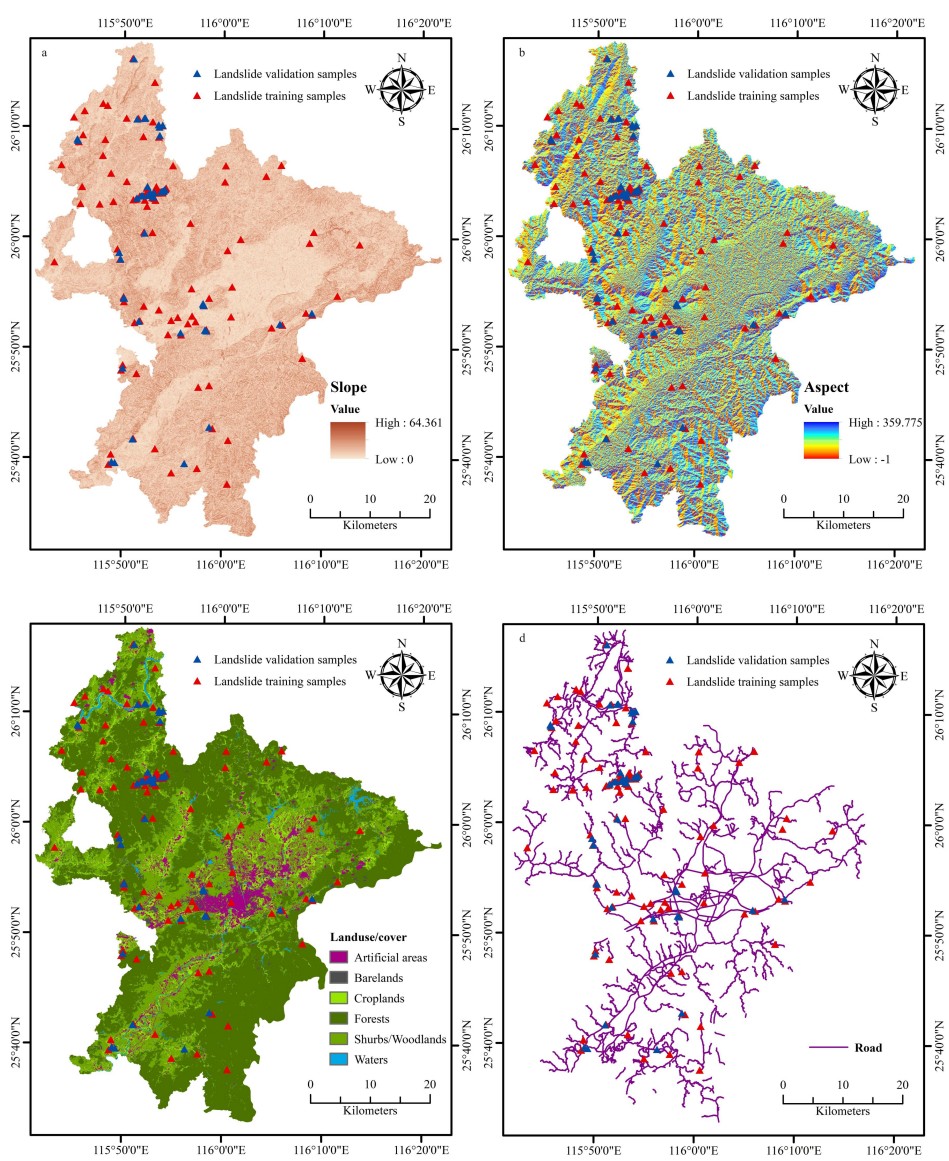

**Fig. 4** Landslide hazard factors: (a) Slope, (b) Aspect, (c) Land use/cover, and (d) Roads.

There were no landslide accidents recorded in the study area caused by earthquakes, so the earthquake factor was not considered in this study.

2.2.1.5 Integrated geo-environmental dataset

The occurrence of landslide is a result of the combined action of all the causative geo-environmental factors (Corominas et al., 2014; Zhu, et al., 2019). In this study, all these factors that may contribute to the occurrence of landslide will be considered for risk modeling. The above processed raster layers namely geological strata, proximity to faults, to lithostratigraphic boundaries, to roads and to to rivers, depth of the weathered crust, soil types and texture, elevation, slope, aspect,

plane curvature, profile curvature, land use/cover, NDVI, multiyear mean annual rainfall, multiyear mean May-July rainfall, multiyear mean March-June rainfall, multiyear mean March-July rainfall were incorporated into a 19-band dataset in Datum WGS 84 and Projection UTM 50 by Layer Stacking function. These raster layers were considered as hazard-causative factors or independent hazard predictors.

2.2.2 Field data

First-hand data were obtained from the field observation during the Geological Hazard Survey Campaign in Ruijin County on a scale of 1/50,000 by the 264 Geological Team of Jiangxi Nuclear Industry in the recent years. The realistic landslides that had occurred in the past ten years were investigated and recorded. Some landslides had been missed during the field observation were digitized on Google Earth as a complement to the former. In total, 155 historical landslide cases were collected. The landslide sites ranged from 20 $m^2$ to 64000 $m^2$ in size. Most of the landslides are small in scale, i.e., less than 900 $m^2$ in study area. To obtain the optimal spatial presentation of samples for RF modeling, the landslide points with areas less than 900 $m^2$ were buffered with radius of 30 m and then rasterized into pixels with size of 30 m (Wu et al., 2018), and those larger than 900 $m^2$, a direct rasterization from vector was conducted. These cases were assigned a value of 1, indicating that the events of landslide have truly taken places, i.e., the probability is 1.

As mentioned above, the non-risk areas (low-slope (< 1-5°) valleys, plain, urban and water-bodies) were integrated into the field dataset as zero-risk area, i.e., the risk probability is 0.

At last, we randomly selected 70 % of the landslide samples (109 cases) plus 70 % of the non-risk zones to constitute a training set (TS) and used the remained ones (45 cases, 30 %) as a validation set (VS).

*2.3 Landslide risk modeling*

Among the machine learning algorithms, Wu et al (2016 and 2018) found that RF and support vector machines (SVM) performed equally well in classification, better than artificial neural networks (ANN), but RF performed best in regression prediction. Hence, RF classification algorithm was selected for geohazard risk modeling in this study. The overall procedure was summarized in Fig. 2 and the detail on modeling, validation and accuracy assessment is given in the following sections.

2.3.1 RF classification of risk probability

RF classification, based on growing decision trees, is an ensemble of tree classifiers that allow to conduct classification by predicting the probability of a given pixel to be classified into the target class through majority voting. The key technique of this algorithm lies in that a bootstrap sample of the TS is used to build each tree, and a stochastic selection of the input variables is searched to determine the best split for each node. Meanwhile, RF algorithm uses out-of-bag (OOB) estimates to define the generalization error and the importance of each variable (Breiman, 2001). RF will not overfit if the number of decision trees (NT) increases to certain level. Thence, NT should be large enough to reduce the OOB error of classification to a stable level in the training process. It is to mention here that instead of classification of land cover types, we employ this algorithm to classify the probability of risk and non-risk for each pixel.


### 2.3.2 Application of RF algorithm

In this study, RF classification was conducted within EnMap-Box which is a package particularly developed to process and analyze image data (Waske et al., 2012). While conducting RF modeling, we kept the combined 19-band dataset as input variables with 109 landslides used as TS and 45 as VS. Some key parameters of RF classification include the Impurity Function, the Stop Criteria (for node splitting), the Number of randomly selected Features (or Number of Variables) at each node and Number of Trees (NT) with classification and regression algorithm (Wu et al., 2016, 2018).

The Gini Coefficient was selected for the Impurity Function. The Stop Criteria was set as the default values which was a Minimum number of samples at a node of 1 or a Minimum impurity calculated based on the Gini index of 0. The Number of randomly selected Features (or Number of Variables) at each node was the square root of all available features. The default value of number of trees was 100 within EnMap-Box. In this study, NT was set to 300 and 500 in order to achieve a better prediction.

After parameterization of RF classification model using the integrated dataset as predictive variables and TS for training and internal validation, the model was applied back to the integrated dataset for landslide prediction, i.e., the probability of landslide occurrence in each pixel. The accuracy of modeling is calculated versus the independent VS.

### 2.3.3 Importance of variables

The importance of variables in the RF classification can be evaluated by variable substitution method. In other words, it can be measured by calculating the difference of the OOB error before and after value substitution. Factor importance of $F_i$ can be expressed as follows:

$$VIM\ (F_i) = \frac{1}{NT} \sum_t errOOB_t^i - errOOB_t \qquad (1)$$

where NT is the number of trees, $errOOB_t$ is an error for tree $t$ of the forests when all the factors are included, $errOOB^i$ refers to an error after removing the factor $F_i$ and $VIM(F_i)$ is variable importance for $F_i$. For RF classification and its result produced, a high value indicates that the factor is of high importance and vice versa.

### 2.3.4 Validation of models

Base on the Confusion Matrix, Precision, Recall, F1 index, Kappa Coefficient (KC) and Overall Accuracy (OA) can be calculated to evaluate the accuracy and performance of landslide risk prediction model (Congalton, 1983).

Generally, both TS and VS can be used to calculate these statistical indices. The evaluation results of TS show the adaptability of the model to the training data, while those of VS reveal the predictive and generalization ability of the model (Tien Bui et al., 2012).

According to previous studies, the smaller the high-risk area predicted by the model, the more historical landslide points concentrated there, which indicates that the model has high predictivity (Cao et al., 2019; Dou et al., 2019). The success rate curve and prediction rate curve can be made respectively by using the landslide risk prediction results of TS and VS. For the study area, the prediction accuracy of landslide risk map can be analyzed and demonstrated by the area under curve (AUC) features (Chung et al., 2003; Yilmaz, 2009; Nicu et al., 2019).




## 3. Results and Discussion

### 3.1 Landslide risk zoning

The landslide risk zoning was achieved based on the modeled risk probability when RF modeling was implemented at NT = 300. All pixels were divided into four levels: low-risk (probability < 0.5), medium-risk (probability ≥ 0.5 and < 0.7), high-risk (probability ≥ 0.7 and < 0.9) and extremely high-risk (probability ≥ 0.9). The landslide risk map of Ruijin was hence produced (Fig. 5a).

The predicted results of the landslide-prone areas were largely consistent with the field survey: (1) High-risk zones were mainly linearly distributed along rivers and roads (Fig. 5b). Many landslides were often caused by river undercutting and artificial cutting for road construction and housing. (2) In the central part of the study area, high-risk zones are concentrated in the Quaternary soil layer, or rather, in the weathered crust, especially along the boundaries of lithologic strata (Fig. 5c). The Quaternary unconsolidated soil layer with loose structure provided rich material for landslides. The boundaries of lithologic strata behaved as unstable structural interfaces, which were important factors for landslides. (3) In the granitic massif, there were also obvious high-risk zones distributed along the roads (Fig. 5b, 5c). Weathering accelerated by humidity, high undulating landform and tectonically active settings of the study area, changed the intrinsic properties of the material and reduced the strength of the near-surface rocks.

Table 1 shows that 24.83 km$^2$ of areas identified as Extremely high risk zones account for 1.02% of the total study area, and High, Medium and Low risk zones take up 227.57 km$^2$ (9.32%), 472.39 km$^2$ (19.36%) and 1715.60 km$^2$ (70.30 %) respectively. Additionally, 94.19% of the field samples, i.e., the realistic landslides, took places in 10.34% of the entire study area, which was categorized as High-risk and Extremely-high risk zones in our risk zone map.

### 3.2 Number of Trees with RF classification

The selection of NT has a great influence on the accuracy of RF modeling. The performance of classification or regression is poor and the error is large when NT is small. As it grows, the OOB error decreases continuously and reaches eventually a threshold (Breiman, 2001). However, the complexity of RF model is directly proportional to NT. If there are too many decision trees, the operating efficiency will decrease as it becomes more time-consuming and the optimal result may not be obtained. The previous study by Wu et al. (2018) confirmed that in both low (e.g., 100) and high NT (e.g., 1000) cases, the algorithm did not perform well, but it did when NT was set to 300 and 500. .It is clear that the OOB error tends to be stable after NT gets greater than 300 (Fig. 6), or rather, the model accuracy becomes greater than 96%. Hence, 300 was finally used for NT when performing landslide risk modeling.

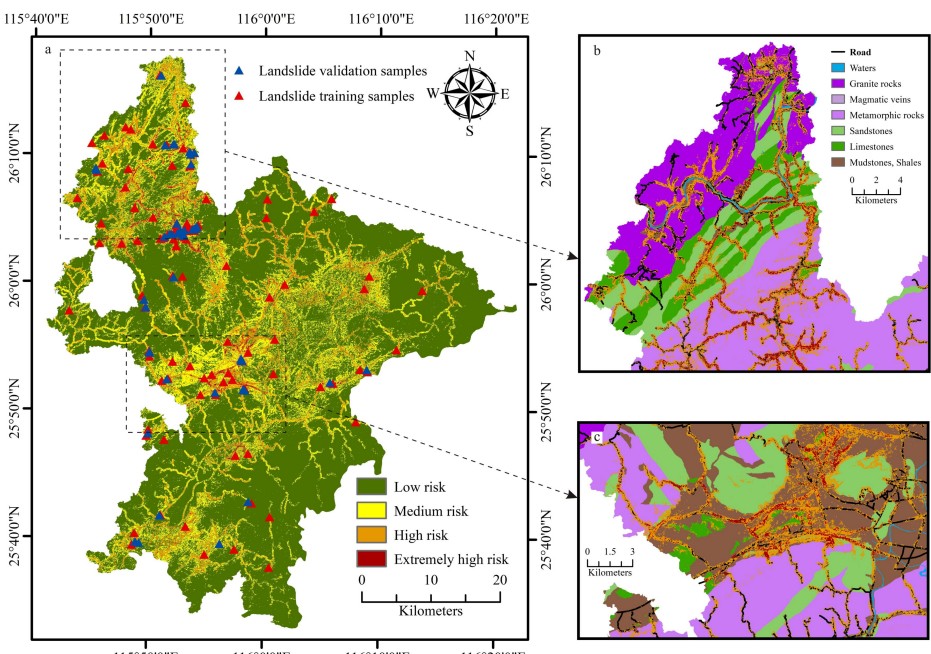

**Fig. 5** The landslide risk zoning map of Ruijin County.

**Table 1.** Distribution of historical landslides within different risk levels

| Landslide Risk Level | Area (km²) | Area Percentages (%) | Number of Historical landslides | Number Percentages (%) |
|---|---|---|---|---|
| Extremely high | 24.83 | 1.02 | 99 | 63.87 |
| High | 227.57 | 9.32 | 47 | 30.32 |
| Medium | 472.39 | 19.36 | 5 | 3.23 |
| Low | 1715.60 | 70.30 | 4 | 2.58 |

*3.3 Importance of hazard factors*

For geohazard assessment, it is critical to understand the importance, more concretely, the role of each geo-environmental factor in such disaster event. In terms of the OBB error, Fig. 7 shows the importance of all the hazard factors considered in this research with the first four factors as follows: 1) distance to road, 2) slope, 3) May-July rainfall, 4) NDVI, 5) elevation, and so on.

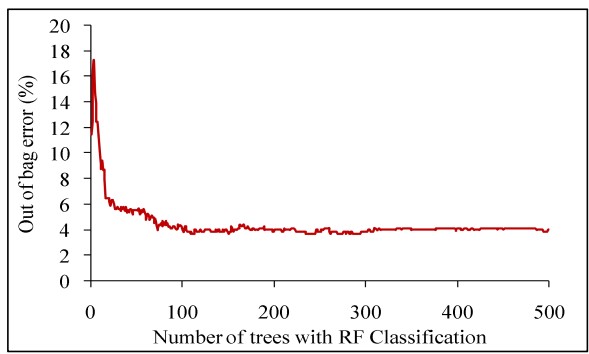

**Fig. 6** Out-of-bag (OOB) error plot versus number of trees (NT) within RF modeling.

In the case of Ruijin, the order of importance seems plausible. Stable slope becomes unstable
as a result of road construction, i.e., slope cutting or housing development, and naturally, steeper
slope has higher propensity to slide due to gravity. May-July rainfall is a triggering factor as it
liquidizes the slippery interface when it reaches certain threshold, i.e., the rainfall amount leading to
saturation of soil after penetration and starting to flow on the soil/rock interface. Actually, the more
rainfall in short time, the higher landslide disaster risk may be developed. Rainfall is thus widely
employed as a Weather Indicator of landslides. NDVI, an autumn mean of five-year period and an
indicator of vegetation abundance, vigor and root system development of forests and woodlands
(herbaceous layer is mostly withered at that time), can largely reflect the stability and instability of
the weathered crust. It is hence reasonable that these factors were identified as the most important
hazard-causative factors in Ruijin though all others may play also a certain role in the geohazard
events.

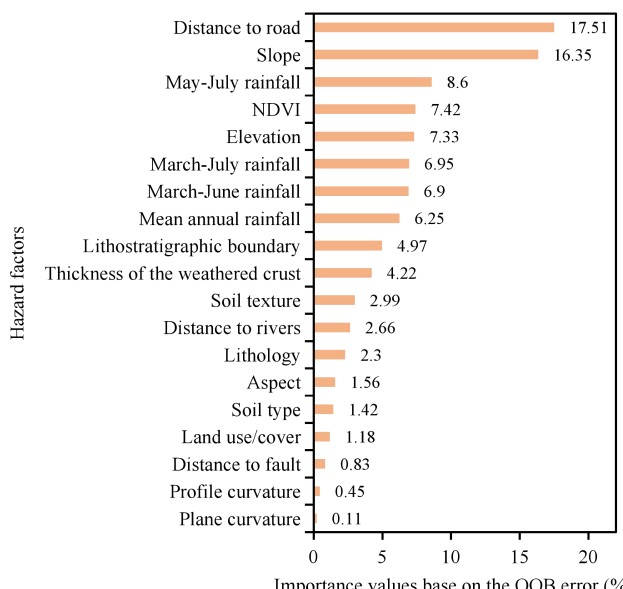

**Fig. 7** Importance of the variables (geo-environmental factors) in provoking landslide events.





The importance of hazard factors associated with landslides had been also discussed by other
authors. The one of Dou et al. (2019) showed that precipitation was the most significant factor but
according to those of Arabameri et al. (2017) and Cao et al. (2019), DEM was the most important
one. Actually, it is understandable that in different geological environments, the mechanism of
landslides may be different and so the importance of hazard factors.
*3.4 Validation of the modeling results*
Only after being validated, has the model potential to be applied elsewhere. We used five
statistical indicators to evaluate the performance of landslide risk model, including Precision, Recall,
F1 Index, KC and OA as mentioned above. Against the training dataset, the Precision, Recall, F1
Index, KC and OA of the modeled results were 0.9908, 0.9818, 0.9863, 0.9724 and 0.9862,
respectively, while they were 0.95, 0.8867, 0.9173, 0.8299 and 0.9149 respectively versus VS
(Table 2). Above all, statistical indicators have shown that the RF classification model has good
performances in predicting the landslide risk.
**Table 2.** Performance of the RF modeling vs training set (TS) and validation set (VS)

| Item | Training Set | Validation Set |
|---|---|---|
| Precision (%) | 99.08 | 95.00 |
| Recall (%) | 98.18 | 88.67 |
| F1 Index (%) | 98.63 | 91.73 |
| KC (%) | 97.24 | 82.99 |
| OA (%) | 98.62 | 91.49 |

NT= 300.

Fig. 8 presented the success rate curve and predicted rate curve versus TS and VS. The AUC
for success rate curve and for prediction rate curve were 0.9936 and 0.9677, respectively. A model
can be considered as appropriate for this kind of risk prediction when the AUC had a value above
0.5 (Chen et al., 2018b; Achour and Pourghasemi, 2019). The landslide risk map had a better
success and prediction rates by RF classification model in this study compared with the results of
other scholars (Chung et al., 2003; Nicu, 2018; Depicker et al., 2020). Hence, RF classification
algorithm has a good predictive capability, and can be extended elsewhere for application.

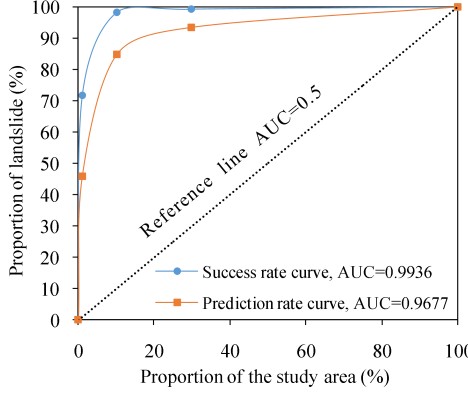

**Fig. 8** Success rate curves and prediction rate curves with associated AUC values.

*3.5 Case verification*
The landslides in the newly-built campus of No. 6 Middle School and the Longzhu Temple
in Ruijin took place very recently. The two new landslide events were predicted as Extremely
high-risk zones in the risk map (Fig. 9a). During the field investigation in July 2019, the middle
school was closed due to this disastrous effect (Fig. 9b); at the Longzhu Temple there were
significant ground bulges along the behind and side wall feet because of the extrusion provoked by
the downward slide of the upper slope (Fig. 9c). Thus, both sites are in danger as landslides
continue gradually.

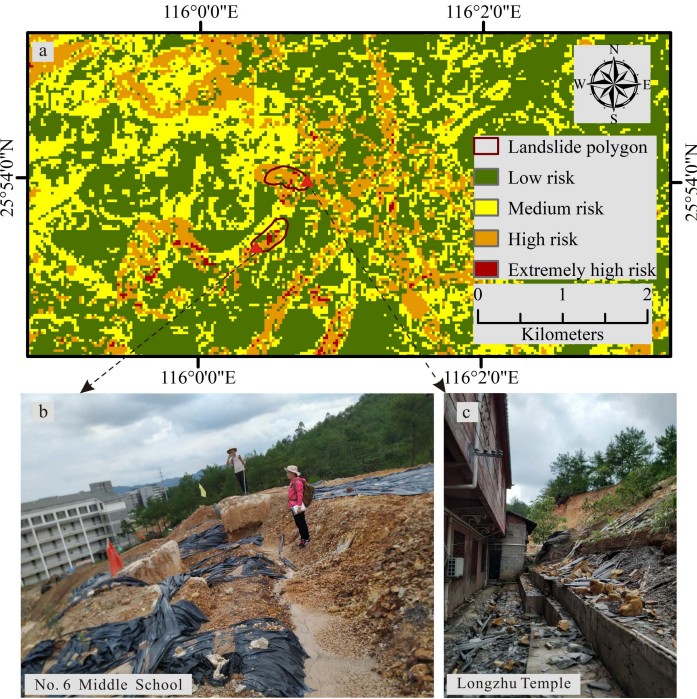

**Fig. 9** The landslide risk zoning map and photographs of landslide in field survey.
**4. Conclusions**
The prediction and prevention of landslide disasters have become essential to secure our
society. This paper presents a research using multi-source geo-environmental dataset as input
variables followed by a RF Classification to model and predict the landslide risk in Ruijin County.
Our results were obtained with high reliability of which OA is 98.62% and 91.49% versus TS and
VS. We believe that our research will be helpful for local government to take action on prevention
and early warning of geohazard to ensure people's safety and property, and to provide theoretical
advice for the infrastructure construction and urban planning.
Our study revealed the critical role of human activity, in particular, road construction, in
landslide events. Most of the observed landslides in Ruijin were actually "man-made". In future
road development we should take its negative impacts into account.



Another innovation lies in finding out a rational digitization and value assignment approach for
non-digital geo-environmental factors such as geological strata, faults, soil, roads and rivers so that
quantitative risk modeling and prediction using RF algorithm can be smoothly achieved.
Our study also illustrates that combination of remote sensing, geological, geomorphic, climatic
and human dimensional data is relevant for such geohazard risk zoning and mapping. RF algorithm
is able to satisfactorily achieve such task. This study can hence serve as a prototype for similar
research elsewhere.

**Acknowledgments**
This research was supported by the Start-up Fund for Scientific Research of the East China University of
Technology, granted to Dr Weicheng Wu (Grant No. DHTP2018001), who is also supported by the Jiangxi
Talent Program, and the Special Innovation Fund for Postgraduate of the East China University of
Technology to Ms Xiaoting Zhou. Field investigation during July, October 2019 was received, the first-hand
field landslide observation data and the 1/50,000 Scale Geological Map of Ruijin were provided by the 264
Geological Team of Jiangxi Nuclear Industry.
**Conflicts of Interest:** The authors declare no conflict of interest.
**Author Contributions:** Conceptualization, X.Z., W.W. and Z.L.; methodology, W.W. and X.Z.; software,
W.W.; validation, X.Z., J.J., M.Z. and Y.L.; formal analysis, X.Z., P.O., W.H. and Y.Z.; investigation, X.Z.,
W.W., Y.S., Z.W., P.O., W.H., Y.Z., L.X. and X.H.; resources, G.Z., R.C., Y.S., Z.W. and T.L.; data curation,
X.Z., P.O., W.H. and Y.Z.; writing—original draft preparation, X.Z. and W.W; writing—review and editing,
W.W.; visualization, F.X. and J.L.; supervision, W.W., Y.Q., S.P., C.S., Y.B., X.Z., X.L., and W.L.; project
administration, W.W. and G.Z.; funding acquisition, W.W. All authors have read and agreed to the published
version of the manuscript.

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
