# Peer review of "Landslide risk zoning in Ruijin, Jiangxi, China"

_Natural Hazards and Earth System Sciences, 2020_

## Referee Comment (RC1) · Qinghan Dong (Referee) · 29 Oct 2020

The paper does address relevant scientific and technical questions within the scope of NHESS. The methodology of using random Forest for regression purposes was clearly described. Although the sample size is relatively small, the results were well presented and sufficient to support the interpretations and the conclusions.

The title and the abstract are pertinent, and the overall presentation is clear and easy to understand to a wide and diversified audience. While the authors gave proper credit to their previous works, the number and quality of the references are appropriate.

The English language is of good quality, fluent, simple and easy to read and understand.

2020-270, 2020.

---

## Author Comment (AC1) · 29 Oct 2020

Dear Dr Dong, Thanks so much for your positive comments. It is well appreciated. Best regards, Weicheng Wu
* * *

---

## Referee Comment (RC2) · Eddy DE PAUW (Referee) · 13 Nov 2020

Overall comments This is a very good paper that illustrates the potential of the Random Forest approach for landslide risk mapping. The research approach is very sound and all factors that affect landslide susceptibility have been taken into consideration. The results are impressive with very high success of the prediction model. The paper is well written, well structured and in good English. Even though the research method relies on the use of high-level statistics, it can be understood by any land resource scientist with only a summary background in statistics. The methodological flowchart is very transparent.

Specific questions/comments: 1) I would suggest to pay attention to the following points: 2) Line 76-77: replace "the" by "important" 3) Line 79: replace "geological and meteorological" by "geological, soil and meteorological" 4) Line 84: replace "land

resource" by "land cover" 5) Lines 85-86: move reference "Qin and Liu (2018)" directly after "ore mineralization" 6) Line 96: "RF algorithm has been rarely applied". Not that rarely apparently, e.g. references Zhang et al. and Pourghasemi and Kerle. 7) Figure 1. I would recommend a wider scale of colors, for instance from green to brown: the range of elevation is from less than 70 m to more than 1200 m. Currently the figure inadequately captures this high range. 8) In Figures 3c, 5b and 5c one does not see any magmatic veins, so why include it in the legend? 9) Given the high range in elevation, I recommend adding a figure with the spatial distribution of either annual rainfall or of the period with the most intensive rainfall. 10) I think it is necessary to indicate the process by which values of the different resource factors (such as river and stream buffers, lithological classes, fault buffers, sand percentages, etc.) were converted into risk scores. I guess by 'expert judgement', but who were the experts? The authors or land users or both groups? 11) Lines 231-232: "forest cover was assigned a value of 1-2" Is it 1 or 2, 1.5 or are there two subclasses of forest cover, one with risk score 1 and another with score 2? 12) Line 236: how are NDVI values converted in risk scores? 13) Areas with slopes <1-5° are considered to be 'non-risk' areas. But a slope angle of 5° is nearly a 10% slope, and that is quite substantial. In the area where I live 15% of all landslides are in the slope class 8-10%. Please confirm that the slope range 1-5° is not too wide, e.g. by noting the presence/absence of actual landslides in that slope range. 14) Could you explain what would have been the consequence of setting NT to 100 instead of 300? Figure 6 indicates that OOB error is already fairly stable at NT=100.

Technical corrections: Lines 134-137: rephrase and simplify the two sentences, as they are currently somewhat confusing. What you want to say is (1) that landslides are more likely on bare land as compared to vegetated areas, (2) slope cuts and excavations for roads and housing exacerbate the risk. Line 194: replace "160-120 m" by"60-120 m". Lines 261 and 353: replace "realistic" by "actual"
* * *
[Figure]

2020-270, 2020.

---

## Referee Comment (RC3) · Anonymous Referee #3 · 14 Nov 2020

The manuscript under review presents a well-structured and clearly readable application of a machine learning (Random Forest)-approach to spatially predict landslide occurence. The illustrations are instructive and well-elaborated.

However, title and scope of the paper are completely misleading. The study just resembles a classification of terrain units (30 m X 30 m pixels) for the probability of landslide occurrence based on several geo-environmental factors and does not consider the temporal probability of such events to occur in the context of a hazard assessment, possibly serving as a basis for landslide risk zonations. The presented analyses have nothing to do with any kind of a risk analysis since no (spatial) vulnerability assessments of potential objects at risk are presented or incorporated in any kind of (spatio-temporal) risk analysis. In such, the paper only resembles the application of a common machine learning approach for landslide susceptibility classification.

[Figure]

Besides this, I am not sure if prediction of landslide susceptibility using any kind of inventory-based analysis is really admissible for such a large territory with only 155 landslides. The landslides are not described at all regarding their typology or triggering mechanisms and their spatial relation to the geo-environmental factors used for susceptibility modelling. The sampling of negatives for modelling is questionable since it is trivial that on shallower slopes landslide susceptibility is low. With such a small landslide data set, negative sampling should be conducted with much greater care on steeper non-landslide slopes to investigate the ability of the method to correctly predict the landslides.

To conclude, the paper adds nothing scientifically new to what is already known from the literature and just represents a case study application that would need much more work to be publishable.

---

## Author Comment (AC2) · 14 Dec 2020

Thank you very much for your detailed comments and constructive suggestions. Here are our reply to what your have raised:

Overall comments This is a very good paper that illustrates the potential of the Random Forest approach for landslide risk mapping. The research approach is very sound and all factors that affect landslide susceptibility have been taken into consideration. The results are impressive with very high success of the prediction model. The paper is well written, well structured and in good English. Even though the research method relies on the use of high-level statistics, it can be understood by any land resource scientist with only a summary background in statistics. The methodological flowchart is very transparent.

Reply: Thank you so much for your positive comments.

[Figure]

Specific questions/comments: 1) I would suggest to pay attention to the following points: 2) Line 76-77: replace "the" by "important"

Reply: Word replaced

3) Line 79: replace "geological and meteorological" by "geological, soil and meteorological"

Reply: Word replaced

4) Line 84: replace "land resource" by "land cover"

Reply: Word replaced

5) Lines 85-86: move reference "Qin and Liu (2018)" directly after "ore mineralization"

Reply: It is done.

6) Line 96: "RF algorithm has been rarely applied". Not that rarely apparently, e.g. references Zhang et al. and Pourghasemi and Kerle.

Reply: It is deleted.

7) Figure 1. I would recommend a wider scale of colors, for instance from green to brown: the range of elevation is from less than 70 m to more than 1200 m. Currently the figure inadequately captures this high range.

Reply: It has been reproduced in the revision as suggested. Thanks.

8) In Figures 3c, 5b and 5c one does not see any magmatic veins, so why include it in the legend?

Reply: The distribution range of the magmatic veins in the study area is relatively small, which cannot be distinguished by the eyes in the figure.

9) Given the high range in elevation, I recommend adding a figure with the spatial distribution of either annual rainfall or of the period with the most intensive rainfall.

Reply: Thanks for comments. The figures with spatial distribution of annual rainfall and March-June rainfall were added (see new Figure 3c, 3d) in the revision.

10) I think it is necessary to indicate the process by which values of the different resource factors (such as river and stream buffers, lithological classes, fault buffers, sand percentages, etc.) were converted into risk scores. I guess by 'expert judgement', but who were the experts? The authors or land users or both groups?

Reply: Thanks for having raised this issue. Resource factors were converted into risk scores by field investigation and expert judgment. It is indicated in the revision.

11) Lines 231-232: "forest cover was assigned a value of 1-2" Is it 1 or 2, 1.5 or are there two subclasses of forest cover, one with risk score 1 and another with score 2?

Reply: Forests have a low proneness, so a value of 1 is assigned. It is modified in the revision.

12) Line 236: how are NDVI converted in risk scores?

Reply: NDVI was produced using late October and early November Landsat images, when crops are harvested and herbaceous vegetation is mostly withered but coniferous and broadleaf trees are still green. We think hence that NDVI represents forest and woodland coverage and vigor. which have . Thus, NDVI value can be directly used to reflect tree density (with developed root systems) that is resistant to landslide event, e.g., high NDVI indicates low risk of landslide given the same other conditions.

13) Areas with slopes <1-5° are considered to be 'non-risk' areas. But a slope angle of 5° is nearly a 10% slope, and that is quite substantial. In the area where I live 15% of all landslides are in the slope class 8-10%. Please confirm that the slope range 1-5° is not too wide, e.g. by noting the presence/absence of actual landslides in that slope range.

Reply: Thank you for reminding. The slope value is 1-3° instead of 1-5°. Here was a mistake. On the basis of field investigation, the "non-risk" areas were selected in

valleys, plains, urban and water-bodies with low hazard proneness.

14) Could you explain what would have been the consequence of setting NT to 100 instead of 300? Figure 6 indicates that OOB error is already fairly stable at NT=100.

Reply: Figure 6 shows that when NT is 100, the OOB error is still in a relatively fluctuating state. As shown in Table 1, the Precision, Recall, F1 Index, Kappa Coefficient and Overall Accuracy against Validation Set (VS) are lower than 300 when NT is set to 100. The Table 1 can be seen in the supplement.

Technical corrections: Lines 134-137: rephrase and simplify the two sentences, as they are currently somewhat confusing. What you want to say is (1) that landslides are more likely on bare land as compared to vegetated areas, (2) slope cuts and excavations for roads and housing exacerbate the risk.

Reply: Thanks for your suggestion. It has been revised and the indication of NDVI clarified.

Line 194: replace "160-120 m" by"60-120 m".

Reply: Thanks for your modification. It is done.

Lines 261 and 353: replace "realistic" by "actual".

Reply: Thanks, it is modified as you suggested.

Please also note the supplement to this comment:
https://nhess.copernicus.org/preprints/nhess-2020-270/nhess-2020-270-AC2-supplement.pdf

---

## Author Comment (AC3) · 14 Dec 2020

We sincerely thank you for the overall feedback and the constructive comments on the manuscript. Below are our responses to what you commented.

The manuscript under review presents a well-structured and clearly readable application of a machine learning (Random Forest)-approach to spatially predict landslide occurence. The illustrations are instructive and well-elaborated.

Reply: Thank you so much for your positive comments.

However, title and scope of the paper are completely misleading. The study just resembles a classification of terrain units (30 m × 30 m pixels) for the probability of landslide occurrence based on several geo-environmental factors and does not consider the temporal probability of such events to occur in the context of a hazard assessment,

possibly serving as a basis for landslide risk zonations.

Reply: Thanks for having raised this issue. This study was aimed at using the Random Forest algorithm to analyze the probability of landslide occurrence and map landslide risk zone of the study area based on a comprehensive consideration of the influences of hazard factors by field investigation and remote sensing technology. Hence, after a careful consideration we decided to use the title of the paper "Landslide risk zoning in Ruijin, Jiangxi, China". As for the temporal probability of such events to occur in the context of a hazard assessment you mentioned, we think it is well worth further investigations. The difficulty encountered was to know the exact occurrence dates and time of the historical landslides in the study area. Even during the field investigation, local people could just tell you "this landslide event took place on day of June or July... " without further concrete information, especially, for those occurred more than 4-5 years ago. Thus, it appears impossible for the time being to analyze the temporal probability of landslide occurrence with high accuracy. Yet, your point will be taken into account in future work when we have more investment to purchase equipment to monitor such events or employ local people to record such information.

The presented analyses have nothing to do with any kind of a risk analysis since no (spatial) vulnerability assessments of potential objects at risk are presented or incorporated in any kind of (spatio-temporal) risk analysis. In such, the paper only resembles the application of a common machine learning approach for landslide susceptibility classification.

Reply: In the study area, the damage caused by landslides mainly happened to the road systems and built up areas around and on slopes (e.g., residential buildings, school and temple), which have been carefully taken into account. The road buffering procedure and propensity weight assignment, were actually something related to the spatial risk analysis. The polygons of residential buildings extracted from Google Earth were superimposed on the landslide risk zoning map to assess the landslide risk of the buildings within 60m and 120 m of spatial extent. It is true that this part was not

well discussed in the original version, but it is added in the revision. Thank you!

Besides this, I am not sure if prediction of landslide susceptibility using any kind of inventory-based analysis is really admissible for such a large territory with only 155 landslides.

Reply: The approach used in our paper is the Random Forest (RF) algorithm which is able to handle the high- and hyper-dimensional datasets with reliable prediction accuracy but requires few training samples (Breiman 2001). This is the big advantage of RF over other algorithms. The accuracy of the RF model versus the verification set (VS), as well as the success rate and prediction rate curve showed good risk prediction results, where Kappa Coefficient (KC) is 0.8299 or 82.99% and the overall accuracy (OA) 91.49%. According to Cohen (1960) and Landis and Koch (1977), this prediction reaches "almost perfect" level. Therefore, despite of the limited amount of landslide points (e.g., 155) for training and modeling, we believe that the risk assessment is reliable. It may hence provide useful reference for risk management prevention in the study area, and the approach be extendable to similar area for landslide prediction and risk assessment. Please check our revision. Thank you!

The landslides are not described at all regarding their typology or triggering mechanisms and their spatial relation to the geo-environmental factors used for susceptibility modelling.

Reply: Thanks for your comments. We are sorry for having overlooked this in the first version. It has been added in the revision, respectively in "2.2.2 Field survey data" a separate subsection "2.3 Distribution of landslide density in each geo-environmental factor" and a new figure 7.

The sampling of negatives for modelling is questionable since it is trivial that on shallower slopes landslide susceptibility is low. With such a small landslide data set, negative sampling should be conducted with much greater care on steeper non-landslide slopes to investigate the ability of the method to correctly predict the landslides.

Reply: Thanks for having raised this issue. Actually, no-risk (or negative as you mentioned) sampling was not conducted in the same way as you had thought. We did this in flat (not shallow) areas, e.g., urban, waters and croplands where slopes are lower than 1-5°, actually, 1-3° supposing that landslide is very unlikely to take places there. For risk modeling, or rather, probability analysis, we have to take samples from two extreme ends, that is, landslide hazard occurred areas (where the probability is 1.0, meaning that the hazards have truly taken places) and no-risk stable areas (whose probability is 0.0). To take no-risk samples on steeper slopes would be a risky issue itself as we were not sure whether such samples were really no-risk ones. Theoretically, any risk modeling shall not involve such uncertainty but just be based on what is sure. Thank you!

To conclude, the paper adds nothing scientifically new to what is already known from the literature and just represents a case study application that would need much more work to be publishable.

Thank you for your general comments, which have revealed the overlooked points in the original version of the paper, and provided us an opportunity to improve it. Some points that need to be clarified are listed here. From a large view, you are right, the paper involved a known machine learning approach, i.e., RF algorithm, and known geo-environmental factors, and appears to have nothing new. But we developed a complete digitization and weight assignment scheme so that no-digital data such as geological map can be digitized with risk propensity indication for risk modeling. Among the tens of documented publications on landslide modeling and prediction, it is rarely seen such a complete and innovative procedure. Secondly, it was after a comparison that we decided to employ RF algorithm as it can process huge volume of hyper-dimensional data for accurate classification and prediction but requires only few samples. This advantage is superior to other machine learning approaches. Thirdly, besides the technical aspect, scientific paper shall provide practical value to our society. In our case, the results can serve as useful reference for local government to

implement disaster reduction and prevention measures. We believe that our revision has been improved and can meet what you had expected. Thanks.

Please also note the supplement to this comment:
https://nhess.copernicus.org/preprints/nhess-2020-270/nhess-2020-270-AC3-supplement.pdf

––––––––––––––––––––––––––